# What Does Softmax Probability Tell Us about Classifiers Ranking Across Diverse Test Conditions?

**Weijie Tu**                                                                *weijie.tu@anu.edu.au*
*Australian National University*

**Weijian Deng**                                                             *weijian.deng@anu.edu.au*
*Australian National University*

**Liang Zheng**                                                              *liang.zheng@anu.edu.au*
*Australian National University*

**Tom Gedeon**                                                               *tom.gedeon@curtin.edu.au*
*Australian National University*
*Curtin University*
*University of Óbuda*

**Reviewed on OpenReview:** *https://openreview.net/forum?id=vtiDUgGjyx*

## Abstract

This work aims to develop a measure that can accurately rank the performance of various classifiers when they are tested on unlabeled data from out-of-distribution (OOD) distributions. We commence by demonstrating that conventional uncertainty metrics, notably the maximum Softmax prediction probability, possess inherent utility in forecasting model generalization across certain OOD contexts. Building on this insight, we introduce a new measure called Softmax Correlation (SoftmaxCorr). It calculates the cosine similarity between a class-class correlation matrix, constructed from Softmax output vectors across an unlabeled test dataset, and a predefined reference matrix that embodies ideal class correlations. A high resemblance of predictions to the reference matrix signals that the model delivers confident and uniform predictions across all categories, reflecting minimal uncertainty and confusion. Through rigorous evaluation across a suite of datasets, including ImageNet, CIFAR-10, and WILDS, we affirm the predictive validity of SoftmaxCorr in accurately forecasting model performance within both in-distribution (ID) and OOD settings. Furthermore, we discuss the limitations of our proposed measure and suggest avenues for future research.

## 1 Introduction

Machine learning (ML) models typically excel on test sets coming from the same distribution as the training set. However, this assumption seldom holds in real-world deployments, where the test environments often experience distribution shifts caused by factors such as sample bias and non-stationarity. Recognizing this challenge, there is a pressing need to assess ML model performance in unlabeled testing environments where traditional evaluation metrics which require data annotations, such as accuracy and $F1$ score, may become infeasible. Hence, our focus shifts towards the vital yet under-explored task of ranking models under these conditions. Specifically, given a pool of trained models and an unlabeled test set, the objective is to efficiently select the most suitable model for utilization.

Various complexity measures have been proposed to predict the in-distribution (ID) accuracy of models (Jiang et al., 2020a; Neyshabur et al., 2015; Bartlett et al., 2017; Keskar et al., 2017; Nagarajan & Kolter, 2019; Neyshabur et al., 2017; Chuang et al., 2021; Jiang et al., 2020b; Smith & Le, 2018; Arora et al., 2018; Dziugaite & Roy, 2017; Dinh et al., 2017). For the OOD test sets, classical domain adaptation theory provides

a partial answer by quantifying the distance between the ID and OOD distributions (Ben-David et al., 2006). Moreover, the "accuracy-on-the-line" phenomenon Miller et al. (2021) shows the linear correlation in the probit scale between ID and OOD performance. However, such phenomenon does not always hold on some distributions (Teney et al., 2022). Building on previous research, our objective is to develop a robust measure capable of ranking models on both ID and OOD datasets *without* testing labels.

Softmax prediction probability has been shown to be useful in analyzing test data in several tasks, such as open-set data detection (Hendrycks & Gimpel, 2016), accuracy estimation (Guillory et al., 2021; Garg et al., 2022), and misclassified input detection (Deng et al., 2022). For example, Hendrycks & Gimpel (2016) and Liang et al. (2018) utilize maximum Softmax prediction probability to identify samples from open-set classes. Driven by these insights, we develop OOD measures based on Softmax probability. Concretely, given various deep models, we aim to develop probability-based measures that monotonically relate to OOD generalization. To validate the feasibility, we conduct extensive and large-scale correlation studies using various models and different types of dataset shifts. We construct a catalog of empirical prediction probability-based measures and create a wide range of experimental setups. We collect 573 different classification models ranging from standard convolutional neural networks to Vision Transformers. We cover 11 ID and OOD datasets with various types of distribution shift, such as ImageNet-V2 (Recht et al., 2019) with dataset reproduction shift and ImageNet-R (Hendrycks et al., 2021a) with style shift.

Based on experimental results, we first show the demand for measures to rank model performance beyond in-distribution accuracy (Miller et al., 2021). Then, we observe that empirical uncertainty measures based on prediction probabilities (*e.g.*, maximum softmax probability) are useful in characterizing OOD generalization to some extent. However, we note that these measures solely account for prediction certainty. In response, we introduce SoftmaxCorr, a method designed to harness both prediction certainty and diversity. Confidence pertains to the certainty of individual predictions, whereas dispersity indicates the spread of predictions across all categories. Specifically, for each classifier, we compute a class-class correlation matrix from all prediction vectors in a test set. Then, we calculate its cosine similarity with a predefined reference matrix, designed to represent desirable prediction patterns, to evaluate whether this classifier makes diverse and certain predictions. The broad correlation study shows the efficacy of SoftmaxCorr in ranking models.

## 2 Related Work

**Predicting generalization in deep learning** studies the ID generalization gap (*i.e.*, the difference between training and test accuracy) of deep neural networks. Representative methods develop complexity measures based on model parameters and training set (Jiang et al., 2020a;b; Neyshabur et al., 2015; Keskar et al., 2017; Bartlett et al., 2017; Neyshabur et al., 2018; Liang et al., 2019; Chuang et al., 2021; Smith & Le, 2018; Arora et al., 2018; Dziugaite & Roy, 2017; Dinh et al., 2017; Dziugaite et al., 2020), such as distance of training weights from initialization (Nagarajan & Kolter, 2019), the product of norms of weights across layers (Neyshabur et al., 2017) and the change of model accuracy with respect to different perturbation levels in training data (Schiff et al., 2021). The above methods assume that training and test data come from the same distribution and do not incorporate characteristics of test data, so we can unlikely make reasonable predictions on a different distribution. To mitigate this limitation, we investigate the model generalization under distribution shift by developing measures that reflect the models' generalization ability on OOD datasets.

**OOD generalization.** Machine learning models should generalize from training distribution to OOD datasets (Djolonga et al., 2021; Koh et al., 2021; Kirsch & Gal, 2022). To study this problem, several benchmarks are proposed (Hendrycks & Dietterich, 2019; Koh et al., 2021; Gulrajani & Lopez-Paz, 2021), such as corruption benchmark (Hendrycks & Dietterich, 2019) and domain generalization testbed (Gulrajani & Lopez-Paz, 2021). Moreover, several methods are proposed to improve model OOD generalization (Volpi et al., 2018; Zhao et al., 2020; Sagawa et al., 2020; Liu et al., 2021; Mansilla et al., 2021; Shi et al., 2021; Krishnamachari et al., 2023; Zhang et al., 2021; Huang et al., 2023; Foret et al., 2020; Huang et al., 2022), such as adversarial domain augmentation (Volpi et al., 2018; Qiao & Peng, 2021; Alhamoud et al., 2022) and inter-domain gradient matching (Shi et al., 2021).

Prior research has established theoretical frameworks for assessing classifier performance amid distribution shifts. Ben-David et al. (2006) introduced the first VC-dimension-based generalization bound, quantifying the difference in classifier error between source and target distributions using a divergence measure. Mansour et al. (2009) extended this analysis to more general loss functions, providing refined generalization bounds via Rademacher complexity. Other studies (Blitzer et al., 2007; Hoffman et al., 2018; Mansour et al., 2008) expanded upon these findings to include multiple source domains. Some works further bound the OOD generalization error based on the divergence between the two distributions (Acuna et al., 2021; Zhang et al., 2019; Tachet des Combes et al., 2020) However, as suggested by Miller et al. (2021), when the distribution shift becomes larger, the above bounds on OOD performance become looser. In addition, Vedantam et al. (2021) report that the adapting theory from domain adaptation is limited in predicting OOD generalization.

**Unsupervised accuracy estimation (UAE)** aims to predict the performance of a given model on various unlabelled out-of-distribution datasets. One line of research utilizes the model outputs on the test sets (Guillory et al., 2021; Garg et al., 2022; Deng et al., 2023; Tu et al., 2023b). For instance, Guillory et al. (2021) uses the maximum value of prediction probability to estimate model accuracy. A parallel line of works predicts model performance by gauging distribution discrepancy between training and test sets using image features (Deng & Zheng, 2021; Tu et al., 2023a; Xie et al., 2024; Lu et al., 2024). For example, Deng & Zheng (2021) uses the first- and second-order statistics of image features and Fréchet Distance to jointly measure the model performance. In contrast, other studies investigate this task through model weights (Yu et al., 2022).

While UAE predicts OOD accuracy of a single model across various test sets, our work focuses on ranking multiple models' OOD performance on an unlabeled test dataset. UAE methods may be deployed in this task by predicting the performance of each single model. However, to derive the estimated performance, most UAE methods (*e.g.*, BoP (Tu et al., 2023a) or Dispersion score (Xie et al., 2024)) require training an accuracy predictor on a number of datasets. These operations make the deployment of methods computational expensive in our tasks. We have included efficient Softmax-based UAE methods in our study

## 3 Task Formulation

**Task definition.** We consider a $K$-way classification task, and let $\mathcal{Y} = \{1, ..., K\}$ denote the label space and $\mathcal{X} \in \mathbb{R}^d$ denote the input space. We are given a labeled training set $\mathcal{D}^S$ that contains data i.i.d drawn from a source distribution $P_S$, and an OOD test set $\mathcal{D}^T := \{(x_i, y_i)\}_{i=1}^N$ that contains $N$ data i.i.d drawn from another distribution $P_T$ ($P_S \neq P_T$). We train $M$ neural network classifiers $\{\phi_m\}_{m=1}^M$ on $\mathcal{D}^S$. Given a sample $(\boldsymbol{x}, y)$ from $\mathcal{D}^T$, the classifier $\phi_m : \mathcal{X} \to \Delta^K$ gives Softmax probabilities for $\boldsymbol{x}$ on $K$ classes, where $\Delta^K$ denote $K-1$ dimensional unit simplex. By testing on $\mathcal{D}^T$, $\phi_m$ yields a prediction matrix $\boldsymbol{P} \in \mathbb{R}^{N \times K}$, whose each row represents prediction probabilities of a test data. Specifically, the prediction matrix satisfies $\sum_{j=1}^K P_{i,j} = 1 \ \forall i \in 1 \ldots N$ and $P_{i,j} \geq 0 \ \forall i \in 1 \ldots N, j \in 1 \ldots K$, where $P_{i,j}$ indicates the probability that $i$-th sample is predicted to the $j$-th class.

The dataset has an evaluation metric (*e.g.*, accuracy) to obtain ground-truth generalization $G_m$ of classifier $\phi_m$. The goal is to design a measure to calculate a score $S_m$ for each classifier $\phi_m$ without access to data annotations. The calculated scores $\{S_m\}_{m=1}^M$ ideally should correlate well with $\{G_m\}_{m=1}^M$, so that we can rank the OOD generalization of models based on the scores.

**Evaluation metrics.** We use Spearman's Rank Correlation coefficient $\rho$ (Kendall, 1948) to measure monotonicity between calculated scores and model generalization. In addition, we also compute the weighted variant of Kendall's rank correlation $\tau_w$, which is shown to be a useful measure when selecting the best-ranked item of interest (You et al., 2021). Both coefficients range from $[-1, 1]$. A value closer to $-1$ or $1$ indicates a strong negative or positive correlation, respectively, and $0$ means no correlation. Similar to Miller et al. (2021) and Baek et al. (2022), we apply the same probit scale to both accuracy and SoftmaxCorr in our experiment for a better linear fit.

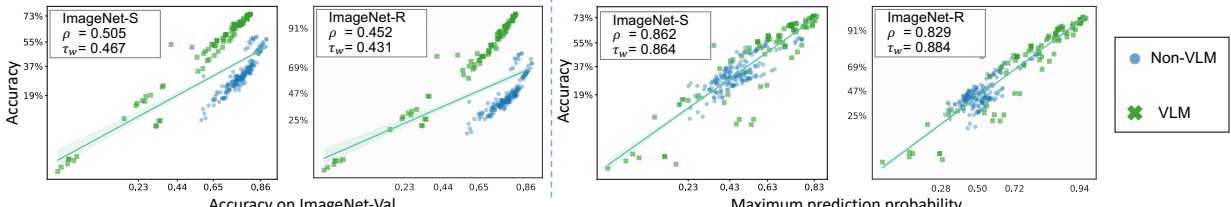

Figure 1: **Correlation study between MaxPred and accuracy (%) on ImageNet-S and ImageNet-R**. Every point denotes a classifier. We use 173 ImageNet models and 89 vision–language models introduced in Section 5. The straight line is fit with robust linear regression (Huber, 2011) and the shadow means the 95% Clopper-Pearson confidence intervals. We show that MaxPred exhibits a moderate correlation with accuracy, while accuracy on ImageNet-validation shows a relatively low correlation with performance on ImageNet-R. Moreover, Vision-Language Models (VLMs) exhibit varying linear trends in terms of their ID and OOD accuracy compared to standard supervised models.

## 4  Softmax Probability-based OOD Measures

### 4.1  What Makes OOD Measures Interesting?

**Beyond Accuracy-on-the-Line (AoL).**  Miller et al. (2021) report an AoL phenomenon where there exists a strong linear correlation between probit-scaled ID and OOD generalization. This implies that ID accuracy is a good predictor of OOD generalization. However, delving deeper into OOD measures is warranted for three compelling reasons. **First**, Miller et al. (2021) discuss that AoL is not universal. That is, on some datasets, ID, and OOD accuracy do not show a clear positive correlation. This point is further discussed by Wenzel et al. (2022). Specifically, Wenzel et al. (2022) suggest two patterns preventing this phenomenon: (1) underspecification (*e.g.*, Camelyon17) where the same ID performance leads to different OOD behavior; (2) models do not transfer information from ID to OOD domains (*e.g.*, DomainNet). That is, despite various ID performance, all models perform poorly on OOD datasets. Paradoxically, identifying the failure patterns itself requires labeled OOD data (Teney et al., 2024).  **Second**, it is demanding and sophisticated to design ID test sets (Engstrom et al., 2020), which are expected to be unbiased and representative of distribution to effectively measure model ID accuracy. Further, it is a trade-off to split a full dataset into training, validation and test sets in terms of training and evaluation quality. **Third**, recent advancements in vision-language models, such as CLIP (Radford et al., 2021) and BLIP (Li et al., 2022), have achieved remarkable zero-shot classification performance. This means that collecting a dataset for a specific task to train such models is not required. Due to their diverse and large-scale training data, it might not be suitable to use a proxy dataset (e.g., ImageNet validation set) to reflect their performance. As shown in Figure 1, Vision-Language models (VLM) exhibit varying linear trends in terms of their ID and OOD accuracy compared to standard supervised models. Furthermore, even among VLMs themselves, consistency in these linear trends is not always guaranteed. These findings suggest that AoL alone does not suffice to accurately rank CLIP models.

**Why Use Softmax Prediction Probability?**  Deep neural networks often exhibit a tendency to provide overly confident predictions, as evidenced by various studies (Ovadia et al., 2019; Minderer et al., 2021; Guo et al., 2017; Hein et al., 2019). Initially, this characteristic might raise doubts about the reliability of using Softmax Prediction as a measure of uncertainty on test data. However, existing research has shed light on its informative nature when analyzing test environments. For instance, Hendrycks & Gimpel (2016) demonstrated that the maximum Softmax prediction probability (MaxPred) for correctly classified samples tends to be higher than that of incorrectly classified or out-of-distribution (OOD) samples. This observation has paved the way for utilizing MaxPred in error/success prediction and ID/OOD detection. Building upon this insight, Guillory et al. (2021) and Garg et al. (2022) have proposed leveraging MaxPred to estimate the accuracy of trained classifiers on test samples.

Moreover, Softmax probabilities can be efficiently computed without requiring any changes to the network architecture or additional data preprocessing. Additionally, they can be derived solely from unlabeled data. These characteristics endow prediction probability-based measures with substantial practical value as reliable

indicators of OOD generalization. Inspired by the above discussion, this work aims to validate the feasibility of using model output-based methods (such as softmax probability) for assessing and ranking models.

**Proof of concept.** The above works imply the prediction probability is likely to be effective in measuring the OOD performance of a pool of models. Given an OOD test set (ImageNet-R) and various ImageNet models, we conduct a correlation study between MaxPred and classification accuracy. In Figure 1, we show that there is a relatively strong correlation between MaxPred and model accuracy ($\rho = 0.829$ and $\tau_w = 0.884$) on ImageNet-R. This indicates that MaxPred is feasible in ranking OOD performance. Based on this observation, we further explore more empirical prediction probability-based measures and develop a more effective measure that exploits more semantics reflected in the prediction probability.

## 4.2 Exploring More Empirical Measures

In addition to accuracy-on-the-line (AoL) and maximum prediction probability (MaxPred), we investigate other empirical measures as follows:

**Average Thresholded Confidence with Maximum Confidence** (ATC-MC) (Garg et al., 2022). This method is used to predict the performance of a specific trained model under distributional shift. Here, we deploy it to rank various models' performance on one particular OOD test set. It firstly identifies a threshold $t$ on $\mathcal{D}^S$ such that the number of samples with confidence score lower than $t$ is equal to the model's error. Then, the ATC on $\mathcal{D}^T$ is given by the number of points whose confidence is less than $t$. The formula is: $ATC = \mathbb{E}_{x \sim \mathcal{D}^T}[\mathbb{I}[\arg\max_{j \in \mathcal{Y}} \mathbf{P}_{:,j} < t]]$, where $\mathbb{I}[E]$ is the binary indicator of event $E$.

**Softmax Gap** (SoftGap). MaxPred only uses the maximal value of Softmax vectors while disregarding values on other entries. Inspired by Baldock et al. (2021), we introduce SoftGap based on MaxPred which further considers the second-largest entry in a prediction vector. Specifically, it calculates the average difference between the largest and second-largest Softmax prediction probabilities over all samples. A high SoftGap indicates a confident prediction, while a low SoftGap suggests confusion between the two classes corresponding to the highest and second-highest probabilities.

## 4.3 Ours: Softmax Correlation (SoftmaxCorr)

**Class-class correlation matrix** Given the prediction matrix $\boldsymbol{P} \in \mathbb{R}^{N \times K}$ predicted by $\phi_m$, a class correlation matrix $\boldsymbol{C} \in \mathbb{R}^{K \times K}$ is computed by $\boldsymbol{C} = \frac{\boldsymbol{P}^\top \boldsymbol{P}}{N}$. An entry $C_{i,j}$ indicates a correlation between prediction probabilities of class $i$ and class $j$ over all samples. We define the sum of diagonals of the correlation matrix as intra-class correlation ($I_a$) and the sum of off-diagonals as inter-class correlation ($I_e$). The sum of $\boldsymbol{C}$ is 1, which means $I_a = 1 - I_e$. Moreover, $trace(\boldsymbol{P}^T \boldsymbol{P}) = \sum_{i=1}^{N} \sum_{j=1}^{K} P_{i,j}^2 = \|\boldsymbol{P}\|^2$, where $\|\cdot\|$ is the Frobenius norm of a matrix. Thus, we can derive $I_a = \frac{\|\boldsymbol{P}\|^2}{N}$. According to Cui et al. (2020), $\|\boldsymbol{P}\|$ has strict opposite monotonicity, and the minimum of the entropy $H(\boldsymbol{P})$ and the maximum of $\|\boldsymbol{P}\|$ could be achieved at the same value. Therefore, $I_a$ reflects the certainty of predictions.

**Motivation:** In domain generalization, it has been observed that class-class correlation encodes class confusion, thereby offering the potential for regularization to enhance model generalization ability (Chen et al., 2022; Jin et al., 2020; Li et al., 2021; Cui et al., 2020). Additionally, existing literature underscores the significance of both high prediction certainty and prediction diversity in identifying discriminative features (Yang et al., 2021; Wang & Isola, 2020; Asano et al., 2019).

Inspired by these insights, we can propose an OOD metric, which leverages a class-class correlation matrix to take into account the two characteristics:

- Prediction certainty: the model's ability to produce confident predictions. Given the strict opposite monotonicity between $I_a$ and entropy $H(\boldsymbol{P})$, prediction certainty can be reflected by a high $I_a$ and consequently a low $I_e = 1 - I_a$ within the class correlation matrix $\boldsymbol{C}$.
- Prediction diversity: relying solely on prediction confidence may be inadequate. For instance, a model may exhibit high certainty yet be biased towards a single class, making the diagonal entries of $\boldsymbol{C}$ skewed.

In such cases, while prediction confidence is high, it does not necessarily indicate high model performance. Driven by this, we further evaluate prediction diversity: each class should be predicted and involved in predictions. To measure this, we expect the diagonals of $C$ to be well-distributed across all classes instead of biased towards few classes.

Combining two characteristics, a desirable class correlation matrix has zeros on the off-diagonal entries, and values on diagonal elements match the distribution of classes. The similarity between the class correlation matrix of an evaluated model and the desirable class correlation matrix reflects the model performance.

**Implementation:** To achieve this, we propose **SoftmaxCorr** and define it as the cosine similarity between the class-class correlation matrix $C$ and a reference matrix $R$: $cos(C, R) = \frac{\sum_{i,j}(C \odot R)_{i,j}}{\|C\| \cdot \|R\|}$, where $\odot$ is the element-wise product between matrices. The reference matrix captures the essence of the desirable class correlations: it is a diagonal matrix whose off-diagonal elements are 0 while its diagonal entries mirror the class marginal distribution. To approximate this class distribution, we use the average prediction probability across test data generated by a zero-shot vision-language model (ViT-bigG/14-CLIPA). The range of the SoftmaxCorr is $[0, 1]$. For a fixed reference matrix, the maximum is obtained 1) when a model confidently assigns probability 1 to the predicted class and 2) the proportion of predictions for each class matches the class distribution. The minimal value is achieved when a model is extremely biased towards one specific class with the prediction probability of one, while the estimated distribution for such a class is zero.

## 5 Experiments

In this section, we first describe three setups including ImageNet, CIFAR-10, and WILDS. Then, we analyze the experiment results of prediction probability-based measures on three setups. After that, we study the impacts of class distribution estimator on predictive ability of SoftmaxCorr. Furthermore, we study whether SoftmaxCorr can rank the performance of model checkpoints along the training trajectory. Also, we validate the effectiveness of SoftmaxCorr under the domain adaption setting. Lastly, we study the correlation between SoftmaxCorr and the accuracy of a single model on various OOD test sets.

### 5.1 Experimental Setup

**ImageNet setup.** We collect 173 models publicly accessible from TIMM (Wightman, 2019). They are trained or fine-tuned on ImageNet (Deng et al., 2009) and have various architectures, training strategies and training paradigms. We use five OOD datasets for the correlation study. Specifically, OOD datasets are: (1) ImageNet-V2 (Recht et al., 2019); (2) ObjectNet (Barbu et al., 2019); (3) ImageNet-S(ketch) (Wang et al., 2019); (4) ImageNet-Blur severity 5 (Hendrycks & Dietterich, 2019); (5) ImageNet-R(endition) (Hendrycks et al., 2021a); ImageNet-R and ObjectNet contain 200 and 113 ImageNet classes respectively.

**CIFAR-10 setup.** We collect 65 networks trained with the scheme provided by Wightman (2017) on CIFAR-10 training set (Krizhevsky et al., 2009). These models have different model architectures. CIFAR-10-Val(idation) is the ID test set. For OOD datasets, we use (1) CIFAR-10.2 (Recht et al., 2018b) (2) CINIC (Darlow et al., 2018) (3) CIFAR-10-Noise with severity 5 (Hendrycks & Dietterich, 2019). We use accuracy as the metric of model generalization.

**WILDS setup.** We consider a classification tasks of this setup: Camelyon17 (Bandi et al., 2018). It is a binary classification dataset where the objective is to classify whether a slide contains a tumor issue. We use 45 models varying in architectures and random seeds. ID and OOD datasets are the default ID validation set and OOD test set respectively. For each task, we follow the same training scheme provided by Koh et al. (2021) to train or fine-tune models.

**Zero-shot vision language models.** In addition to models which are trained on the ID training dataset, we also consider 89 zero-shot vision-language models, including CLIP (Radford et al., 2021), SigLIT (Zhai et al., 2023), BLIP (Li et al., 2022), BLIP-2 (Li et al., 2023) and Flava (Singh et al., 2022). We use default

| Setup | Dataset | Validation Required | | | | Validation Free | | | | | |
| | | AoL | | ATC-MC | | MaxPred | | SoftGap | | SoftmaxCorr | |
| | | $\rho$ | $\tau_w$ | $\rho$ | $\tau_w$ | $\rho$ | $\tau_w$ | $\rho$ | $\tau_w$ | $\rho$ | $\tau_w$ |
| ImageNet | ImageNet-V2 | 0.954 | 0.911 | 0.994 | 0.961 | 0.711 | 0.597 | 0.796 | 0.644 | 0.921 | 0.758 |
| | ImageNet-A | 0.593 | 0.636 | 0.830 | 0.895 | 0.756 | 0.813 | 0.805 | 0.846 | 0.964 | 0.915 |
| | ImageNet-R | 0.452 | 0.431 | 0.950 | 0.954 | 0.829 | 0.884 | 0.902 | 0.911 | 0.951 | 0.928 |
| | ImageNet-S | 0.505 | 0.467 | 0.981 | 0.959 | 0.862 | 0.864 | 0.887 | 0.871 | 0.935 | 0.909 |
| | ObjectNet | 0.619 | 0.545 | 0.961 | 0.821 | 0.883 | 0.849 | 0.908 | 0.865 | 0.963 | 0.895 |
| | ImageNet-Blur | 0.637 | 0.576 | 0.937 | 0.905 | 0.816 | 0.821 | 0.844 | 0.845 | 0.961 | 0.907 |
| | Average | 0.627 | 0.594 | 0.942 | **0.916** | 0.810 | 0.805 | 0.857 | 0.830 | **0.949** | 0.885 |
| CIFAR-10 | CIFAR-10.2 | 0.983 | 0.949 | 0.992 | 0.966 | 0.837 | 0.809 | 0.854 | 0.818 | 0.894 | 0.836 |
| | CINIC | 0.866 | 0.890 | 0.952 | 0.902 | 0.665 | 0.663 | 0.690 | 0.681 | 0.821 | 0.763 |
| | CIFAR-10-Noise | 0.641 | 0.765 | 0.219 | 0.461 | 0.051 | 0.131 | 0.139 | 0.213 | 0.931 | 0.917 |
| | Average | 0.830 | **0.868** | 0.721 | 0.776 | 0.518 | 0.534 | 0.561 | 0.571 | **0.892** | 0.846 |
| WILDS | Camelyon17-OOD | −0.021 | −0.072 | −0.111 | −0.075 | 0.192 | 0.320 | 0.192 | 0.320 | 0.420 | 0.560 |
| | DomainNet-OOD | 0.350 | 0.219 | 0.513 | 0.254 | 0.403 | 0.274 | 0.407 | 0.258 | 0.740 | 0.680 |
| | Average | 0.165 | 0.074 | 0.201 | 0.217 | 0.298 | 0.597 | 0.300 | 0.289 | **0.580** | **0.620** |
| Average over three setups | | 0.598 | 0.575 | 0.747 | 0.751 | 0.637 | 0.693 | 0.675 | 0.609 | **0.864** | **0.824** |

Table 1: **Method comparison on ImageNet, CIFAR-10, WILDS and DomainNet**. We compare SoftmaxCorr with four measures: accuracy-on-the-line (AoL) (Miller et al., 2021), average thresholded confidence with maximum confidence (ATC-MC) (Garg et al., 2022), MaxPred (Hendrycks & Gimpel, 2016) and SoftGap (Baldock et al., 2021). Spearman's rank correlation ($\rho$) and weighted Kendall's correlation ($\tau_w$) are metrics. The highest correlation in each row is highlighted in **bold** and the second highest is in blue. We show that our method is stable and yields the highest average correlations over three setups.

prompt set for corresponding models. If the default prompt sets are not provided, "`A picture of {class}.`" is deployed. Unless specified, we use ViT-bigG/14-CLIPA to estimate class distribution for all setups.

## 5.2 Main Observations

**Albeit requiring no access to validation/proxy set, SoftmaxCorr exhibits a strong correlation with model generalization.** In Figure 2 and Table 1, we observe that SoftmaxCorr is indicative of model performance under the three setups. Particularly on the ImageNet setup, SoftmaxCorr has consistently strong correlations with models' OOD performance. For example, the average Spearman's Rank Correlation coefficient $\rho$ is 0.949, 0.627, 0.942, 0.810, and 0.857 for our method, AoL, ATC-MC, MaxPred and SoftGap, respectively. We also notice that AoL exhibits very strong correlation with model accuracy on ImageNet-V2 ($\rho = 0.954$), but such a correlation drops noticeably on other OOD datasets. This implies that it is insufficient to rank both supervised models and vision–language models based solely on this phenomenon. On CIFAR-10 and WILDS, while on some test sets it does not present the strongest correlation, we still think that SoftmaxCorr is a preferred measure because it has very competitive average correlation scores. For Camelyon17, we see that all methods are less useful. We speculate this is caused by the under-specification phenomenon (Wenzel et al., 2022) where the model relies on spurious features.

**SoftmaxCorr gives stable correlation, while the other four measures have mixed performance on different test sets.** On CIFAR-10-Noise, we find that SoftmaxCorr correlates well with model performance ($\rho = 0.931$ and $\tau_w = 0.917$). In contrast, AoL, ATC-MC, MaxPred, SoftGap show a weaker correlation. Although ATC-MC has a higher average weighted Kendall's correlation than SoftmaxCorr ($\rho = 0.916$ *vs.* 0.885) on ImageNet setup, it exhibits a weaker correlation on CIFAR-10 and WILDS setups. While in some cases SoftmaxCorr does not present the highest correlation, we emphasize that it overall gives more stable and stronger correlations. Thus, we think SoftmaxCorr is generally indicative of model generalization.

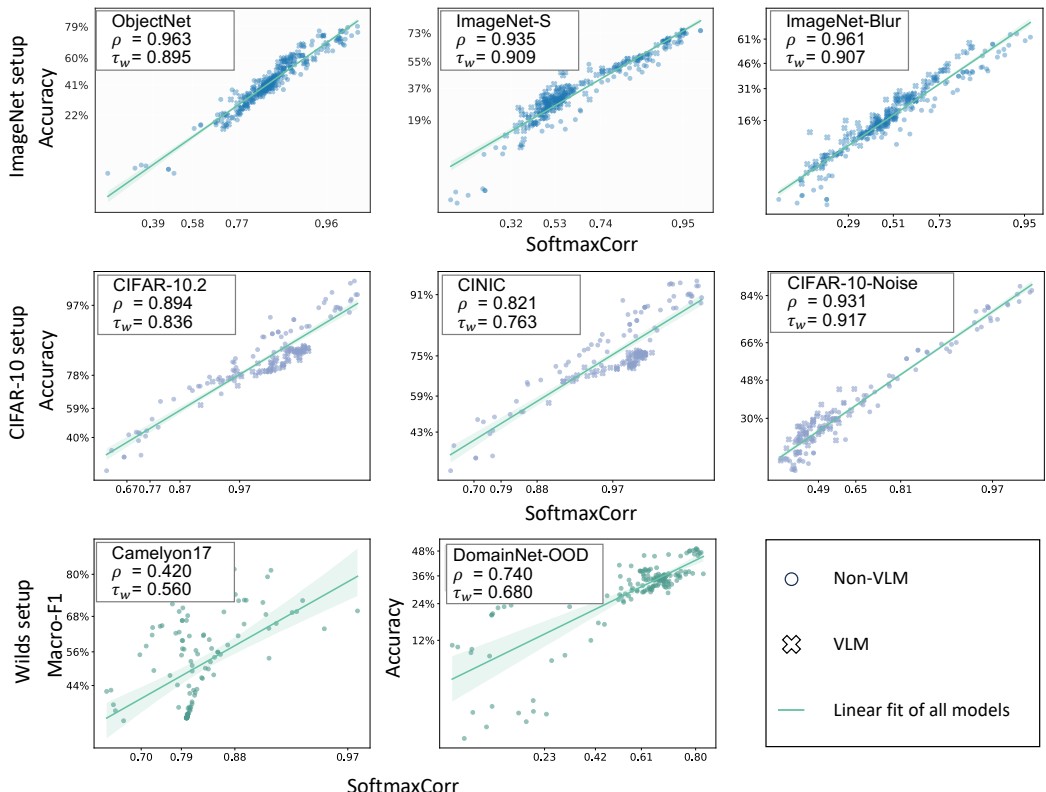

Figure 2: **SoftmaxCorr *vs.* model generalization under ImageNet, CIFAR-10 and WILDS setups**. In each subfigure, each point denotes a model trained for the corresponding task. For ImageNet setup, OOD test sets are ObjectNet and ImageNet-S and ImageNet-Blur. For CIFAR-10 setup, OOD test sets are CIFAR-10.2, CINIC and CIFAR-10-Noise. For WILDS, OOD test sets are Camelyon17-OOD and DomainNet-OOD. The *y*-axis is top-1 accuracy, top-1 accuracy and macro-F1 for the three setups, respectively. Straight lines are fit with robust linear regression (Huber, 2011). Axes are probit scaled as described in Section 3. We observe that SoftmaxCorr is a reliable and effective metric. Particularly on ImageNet, SoftmaxCorr is predictive of model generalization with strong performance ($\rho > 0.92$).

**Compared to MaxPred and SoftGap, SoftmaxCorr better utilizes Softmax prediction probabilities.** We use SoftGap on top of MaxPred as a simple approach to explicitly consider more entries (the second largest probability) in Softmax predictions. As shown in Table 1, SoftmaxCorr and SoftGap both achieve higher correlation results than MaxPred ($\rho = 0.864, 0.675$ and $0.637$, respectively). This indicates that it is helpful to analyze the distribution of softmax outputs. Compared with MaxPred and SoftGap, the class-wise correlation considered in our SoftmaxCorr better reveals the knowledge encoded by Softmax predictions. This is supported by the higher average correlation and more stable performance of SoftmaxCorr.

### 5.3 Sensitivity Analysis on Test Set Size

We study the sensitivity of SoftmaxCorr to test set size. Specifically, we reduce the dataset size by randomly sampling $1\%, 5\%, 10\%$ and $30\%$ of the original data. We report the averaged Spearman's correlation of three random runs on six datasets (*e.g.*, ImageNet-V2, ImageNet-R, ObjectNet, CINIC, CIFAR-10-Noise and DomainNet-OOD). As shown in Table 3, we observe that when the number of test data is very small (1%), the correlation of SoftmaxCorr drops. When the dataset size increases ($\geq 10\%$), SoftmaxCorr exhibits a stable and high correlation with model performance. This suggests that SoftmaxCorr requires a reasonable number of samples to capture model generalization.

| Dataset | Certainty | Diversity | SoftmaxCorr |
|---|---|---|---|
| ImageNet-V2-A | 0.619 | 0.551 | **0.921** |
| ImageNet-R | 0.851 | 0.743 | **0.951** |
| ObjectNet | 0.881 | 0.812 | **0.963** |
| CIFAR-10.2 | 0.881 | 0.766 | **0.894** |
| CINIC | 0.643 | 0.543 | **0.821** |
| DomainNet-OOD | 0.412 | 0.139 | **0.740** |

Table 2: Comparison between SoftmaxCorr and two variants (Certainty and Diversity). Rank correlation ($\rho$) is used as the metric.

| Dataset | 1% | 5% | 10% | 30% | 100% |
|---|---|---|---|---|---|
| ImageNet-V2 | 0.851 | 0.791 | 0.817 | 0.862 | 0.921 |
| ImageNet-R | 0.913 | 0.949 | 0.945 | 0.951 | 0.951 |
| ObjectNet | 0.906 | 0.950 | 0.955 | 0.963 | 0.963 |
| CIFAR-10-Noise | 0.910 | 0.933 | 0.930 | 0.930 | 0.931 |
| CINIC | 0.596 | 0.838 | 0.837 | 0.832 | 0.821 |
| DomainNet-OOD | 0.758 | 0.784 | 0.789 | 0.789 | 0.740 |

Table 3: **Sensitivity analysis of SoftmaxCorr on test set sizes**. We test four sampling ratios and report $\rho$ on six datasets. SoftmaxCorr is stable given a reasonable number of samples.

| Method | ImageNet-V2-A | ImageNet-R | ObjectNet | CIFAR-10.2 | CINIC | DomainNet-OOD |
|---|---|---|---|---|---|---|
| Disagreement | $-0.379$ | 0.746 | 0.021 | 0.264 | 0.314 | $-0.083$ |
| SoftmaxCorr | 0.921 | 0.951 | 0.963 | 0.894 | 0.821 | 0.740 |

Table 4: Comparison between SoftmaxCorr and Disagreement. Rank correlation ($\rho$) is used as the metric.

### 5.4 Impacts of Class Distribution Estimator

In previous experiments, we solely use a vision-language model as the reference model to estimate class distribution. Another baseline usage of such a reference model is to measure the disagreements between the predictions of evaluated models and the reference model (Disagreement). In Table 4, we compare Spearman's rank correlation of Disagreement and SoftmaxCorr. We observe that SoftmaxCorr consistently gives a stronger correlation than Disagreement. Disagreement explicitly measures the similarity between the evaluated models and the reference model, so the performance of the reference model significantly influences the predictive ability of Disagreement. In contrast, SoftmaxCorr only uses it for estimating the empirical class distribution, which constitutes the diagonal elements of the reference matrix $\boldsymbol{R}$, with all off-diagonal elements set to zero.

To study the influence caused by the estimator, we further use a less accurate zero-shot vision-language model (ViT-H-14) and an ideal estimator for calculating the marginal class distribution. We evaluate them on ObjectNet, because it has imbalanced class distribution. In Figure 3, we observe that with a moderately accurate ViT-H-14, SoftmaxCorr remains predictive, and the reference estimator further enhances SoftmaxCorr. Note that, we only use the average of softmax prediction probability to estimate class weights. Some better-designed algorithms (Lipton et al., 2018; Garg et al., 2020; Sun et al., 2022) may improve SoftmaxCorr's stability, and we leave it as future work.

### 5.5 Different Characterizations of Correlation Matrix

To investigate the importance of prediction diversity and certainty for predicting OOD generalization, we compare SoftmaxCorr with two variants: (1) the sum of diagonal entries in the class correlation matrix (Certainty). It measures whether the values in diagonal entries are large, indicating prediction certainty; (2) the Euclidean distance between diagonal elements in the class correlation matrix and estimated class distribution (Diversity). It measures whether models make diverse predictions whose distribution matches the estimated distribution of each class. In Table 2, we present Spearman's rank correlation of three methods on six datasets from three setups. Both variants give weaker correlation strength than SoftmaxCorr. Specifically, SoftmaxCorr is more predictive of OOD generalization than Certainty and Diversity ($\rho = 0.821$ *vs.* 0.643 *vs.* 0.543) on CINIC. This indicates that it is important to measure both prediction diversity and certainty for OOD generalization assessment.

### 5.6 Effectiveness of SoftmaxCorr for Domain Adaptation

On ImageNet, CIFAR and WILDS setups, all models are trained by standard empirical risk minimization and do not use the unlabeled OOD samples from training. In some scenarios, domain adaptation (DA) algorithms

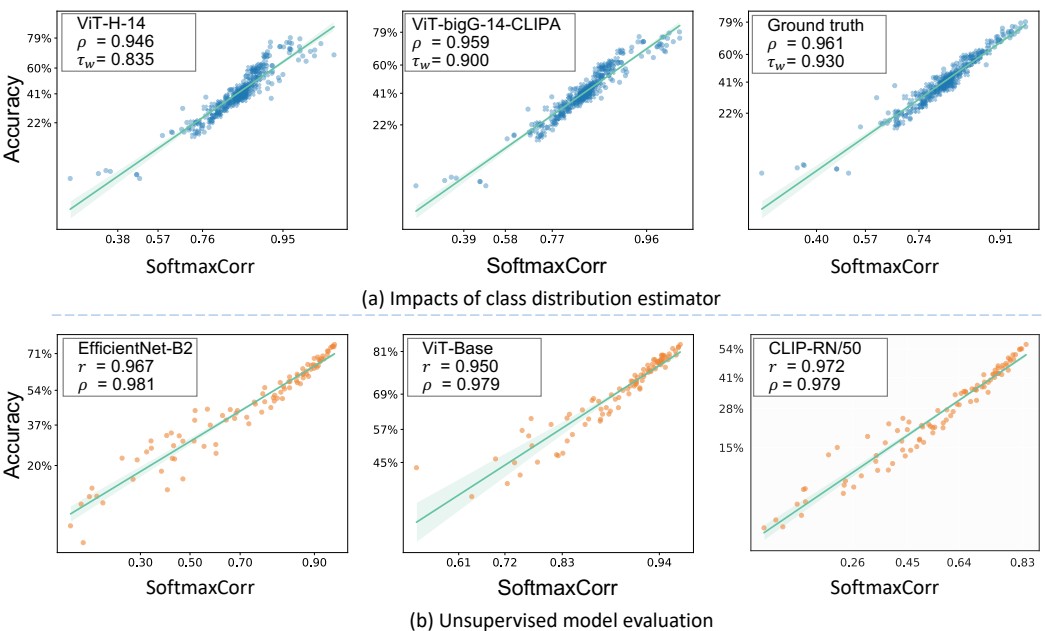

(a) Impacts of class distribution estimator

(b) Unsupervised model evaluation

Figure 3: **(a) Impacts of class distribution estimator**, we use three estimators: ViT-H-14, ViT-bigG-14-CLIPA and the ground truth. We find SoftmaxCorr is fairly stable. **(b) SoftmaxCorr *v.s.* accuracy on ImageNet-C benchmark**. In every subfigure, each dot indicates a dataset of ImageNet-C.. We see strong correlations between SoftmaxCorr and OOD accuracy on various test set.

are employed for learning target-adaptive models with additional unlabeled OOD samples (Kouw & Loog, 2019; Zhou et al., 2022). To explore whether SoftmaxCorr is still effective in assessing the generalization of these models, we conduct a correlation study under DomainNet setup (Peng et al., 2019; Sagawa et al., 2021). The models are trained by 9 different DA algorithms (*e.g.*, DeepCORAL (Sun & Saenko, 2016), DANN (Ganin et al., 2016)). In Table 1, we observe that SoftmaxCorr performs reasonably on DomainNet-OOD. We also notice that ID accuracy-based methods (AoL and ATC-MC) become less useful. DA algorithms likely focus on improving OOD performance, while ID accuracy may not be enhanced accordingly.

## 5.7 SoftmaxCorr Reflects a Model's Generalization on Various OOD Test Sets

We investigate how a given trained model generalizes to various OOD datasets. In detail, we evaluate a single model on all test sets of ImageNet-C benchmark (Hendrycks & Dietterich, 2019) and conduct a correlation study between accuracy and SoftmaxCorr. We additionally use Pearson's correlation ($r$) to measure the overall linear trend. This coefficient varies in $[-1, 1]$. A value closer to $-1/1$ indicates better negative/positive linearity and 0 means no correlation. We use ImageNet networks: EfficientNet-B2 (Tan & Le, 2019), ViT-Base (Dosovitskiy et al., 2020) and CLIP-RN/50 (Radford et al., 2021). Figure 3 shows a strong linear relationship and high-rank correlation ($r > 0.95$ and $\rho > 0.97$). It indicates that with linear regression, SoftmaxCorr can also help estimate the accuracy of a given model on various test sets.

## 5.8 Evaluation Along Training Trajectory

In previous sections, SoftmaxCorr is used to measure the performance of models varying in different architectures and training strategies. In practice, we are sometimes interested in evaluating the models at different training checkpoints. Hence, we analyze whether SoftmaxCorr is helpful in this case. We collect prediction probabilities on CINIC every 10 epochs along the training process of ResNet-20, DenseNet-121 (Huang et al., 2017), VGG11 (Simonyan & Zisserman, 2014) and MobileNet (Howard et al., 2017) trained on CIFAR-10. In Fig. 4, we observe SoftmaxCorr has a high-rank correlation ($\rho > 0.93$) with model performance. This means we can potentially apply SoftmaxCorr to assay model generalization along the training process.

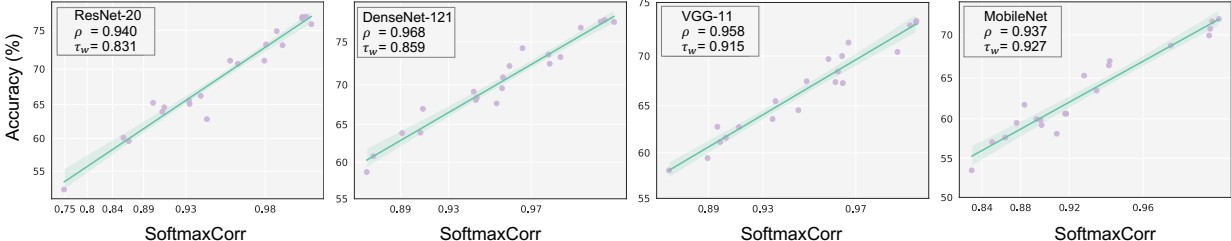

Figure 4: **Correlation analysis: SoftmaxCorr and accuracy on CINIC**. Each point represents a checkpoint. We consider CIFAR-10 models: ResNet-20, DenseNet-121, VGG-11 and MobileNet. Axes are probit scaled as in Section 3. In each subfigure, every point means a checkpoint of the model along the training process. For four models, we see strong correlations ($\rho > 0.93$). This suggests that SoftmaxCorr is helpful in assessing checkpoints along the training process.

## 6   Discussion and Potential Directions

**Discussion on imbalanced test sets.**   SoftmaxCorr is defined as cosine similarity between correlation matrix $C$ and a reference matrix $R_K$. The reference matrix is a diagonal matrix and each entry represents the corresponding class distribution. To derive class distribution, a zero-shot vision-language model is used for efficient deployment without training and shows strong robustness towards distribution shifts. Consequently, the predictive ability of SoftmaxCorr correlates to the deployed vision-language model and the method used to estimate the distribution. Hence, it would be beneficial to use advanced label shift estimation techniques (Lipton et al., 2018; Garg et al., 2020; Sun et al., 2022) and employ more performant vision-language models, and we leave it as future work.

**Potential OOD measures.**   This work proposes SoftmaxCorr to use class-wise relationships encoded by Softmax prediction probabilities. Here, we discuss other potential ways. **First**, SoftGap computes the difference between the largest and second-largest prediction probabilities. We show that SoftGap exhibits a stronger correlation with performance compared to MaxPred. It would be interesting to improve SoftGap by utilizing more probabilities (*e.g.*, top five probabilities). **Second**, for a perfectly calibrated model, its MaxPred over a test set corresponds to its accuracy. Yet, calibration methods seldom exhibit desired performance under distribution shift (Ovadia et al., 2019). That said, it would be promising to study post-hoc calibration methods for OOD datasets, which benefits MaxPred for assessing model generalization. **Last**, this work focuses on Softmax prediction probability. We tested our method based on logits but no obvious correlation is exhibited. This may be because the logits of different models vary in significantly different ranges. We also think that studying other model statistics (*e.g.*, weights and feature representations) would be interesting.

## 7   Conclusion

This paper studies an important problem of assaying and ranking model generalization under distribution shifts. To this end, we explore the use of Softmax prediction probability for developing OOD measures. We start by identifying the demand for OOD measures beyond accuracy-on-the-line and finding that maximum Softmax prediction probability is to some extent useful to measure the OOD performance. We then propose Softmax Correlation (SoftmaxCorr) which leverages class confusion encoded by the class-class correlation matrix and thus better reflects the overall quality of the classifier predictions. To validate the usefulness of SoftmaxCorr, we compare it with four other empirical measures across 11 datasets under ImageNet, CIFAR-10 and WILDS setups. We observe SoftmaxCorr generally presents a stable and high correlation with model performance on different OOD datasets. This paper establishes some baseline usage of Softmax prediction probability and a specific improvement, and more investigation will be made in the future.

### Acknowledgements

We sincerely thank all the anonymous reviewers and area chairs for their constructive comments and valuable suggestions, which have greatly helped in enhancing this paper.

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

# A Appendix

In the appendix, we first introduce the experimental details including access to models, datasets, and computation resources.

## A.1 Experimental Setup

### A.1.1 Datasets

We carefully check the licenses of all datasets used in the experiment and list the open sources to them.

`ImageNet` (Deng et al., 2009):
(https://www.image-net.org);
`ImageNet-C` (Hendrycks & Dietterich, 2019):
(https://github.com/hendrycks/robustness);
`ImageNet-V2` (Recht et al., 2019):
(https://github.com/modestyachts/ImageNetV2);
`ImageNet-Adversarial` (Hendrycks et al., 2021b):
(https://github.com/hendrycks/natural-adv-examples);
`ImageNet-Rendition` (Hendrycks et al., 2021a):
(https://github.com/hendrycks/robustness);
`ImageNet-Sketch` (Wang et al., 2019):
(https://github.com/HaohanWang/ImageNet-Sketch);
`ObjectNet` (Barbu et al., 2019):
(https://objectnet.dev/download.html);
`CIFAR-10` (Krizhevsky et al., 2009):
(https://www.cs.toronto.edu/ kriz/cifar.html);
`CIFAR-10-C` (Hendrycks & Dietterich, 2019):
(https://github.com/hendrycks/robustness);
`CIFAR-10.2` (Recht et al., 2018a):
(https://github.com/modestyachts/CIFAR-10.1);
`CINIC` (Darlow et al., 2018):
(https://www.v7labs.com/open-datasets/cinic-10);
`Camelyon17` (Bandi et al., 2018):
(https://camelyon17.grand-challenge.org/);
`DomainNet` (Peng et al., 2019):
(http://ai.bu.edu/M3SDA/);

### A.1.2 Model Pool

**ImageNet models** are publicly available via TIMM (Wightman, 2019). Models pre-trained by contrastive learning can be used from model-vs-human project (Geirhos et al., 2021). Some MAE models (He et al., 2022) are accessible from (https://github.com/facebookresearch/mae).
PIRL
InsDis
MoCo
MoCoV2
InfoMin
simclr_resnet50x4
simclr_resnet50x1
simclr_resnet50x2
resnet50_l2_eps1
resnet50_l2_eps0_5
resnet50_l2_eps0_25

```
resnet50_l2_eps0_03
resnet50_l2_eps0_01
ens_adv_inception_resnet_v2
adv_inception_v3
tf_efficientnet_b0_ap
resnet50_trained_on_SIN_and_IN
resnet50_trained_on_SIN_ and_IN_then_finetuned_on_IN
resnet50_trained_on_SIN
resnet50_augmix
resnet50_cutmix
resnet50_deepaugment
resnet50_deepaugment_and_augmix
resnet50_feature_cutmix
resnet50_pixmix
convnext_xlarge_384_in22ft1k
convnext_xlarge_in22ft1k
resnetv2_152x2_bitm
resnetv2_50x1_bitm
convnext_base_384_in22ft1k
convnext_base_in22ft1k
resmlp_big_24_224_in22ft1k
resmlp_big_24_distilled_224
tf_efficientnetv2_s_in21ft1k
tf_efficientnetv2_m_in21ft1k
tf_efficientnetv2_l_in21ft1k
tf_efficientnetv2_xl_in21ft1k
vit_large_patch16_384
swin_large_patch4_window12_384
beit_large_patch16_512
beit_large_patch16_384
beit_large_patch16_224
beit_base_patch16_384
vit_base_patch16_384
vit_small_r26_s32_384
vit_tiny_patch16_384
vit_large_r50_s32_384
mixer_b16_224_miil
resmlp_big_24_224
resnetv2_50x1_bit_distilled
ig_resnext101_32x16d
ig_resnext101_32x32d
ig_resnext101_32x8d
ig_resnext101_32x48d
resnext101_32x16d_wsl
tf_efficientnet_l2_ns_475
tf_efficientnet_b7_ns
tf_efficientnet_b6_ns
tf_efficientnet_b5_ns
ssl_resnext101_32x8d
ssl_resnext101_32x16d
swsl_resnext101_32x8d
swsl_resnext101_32x16d
ssl_resnext101_32x4d
ssl_resnext50_32x4d
```

```
ssl_resnet50
swsl_resnext101_32x4d
swsl_resnext50_32x4d
swsl_resnet50
swin_small_patch4_window7_224
swin_base_patch4_window12_384
deit_base_patch16_224
deit_small_distilled_patch16_224
resmlp_36_224
cait_s36_384
cait_s24_224
convit_base
convit_tiny
twins_pcpvt_base
eca_nfnet_l1
xcit_tiny_24_p8_384_dist
efficientnet_b1
efficientnet_b3
efficientnet_b4
tf_efficientnet_b2
tf_efficientnet_lite1
convnext_base
convnext_small
resnetrs350
pit_xs_distilled_224
crossvit_small_240
botnet26t_256
tinynet_e
tinynet_d
repvgg_b2g4
mnasnet_small
dla46x_c
lcnet_050
tv_resnet34
tv_resnet50
tv_resnet101
tv_resnet152
densenet121
inception_v4
resnet26d
mobilenetv2_140
hrnet_w40
xception
xception41
resnet18
resnet34
seresnet50
mobilenetv2_050
seresnet33ts
wide_resnet50_2
wide_resnet101_2
resnet18d
hrnet_w18_small
gluon_resnet152_v1d
```

```
hrnet_w48
hrnet_w44
repvgg_b2
densenet201
hrnet_w18_small
resnet101d
gluon_resnet101_v1d
gluon_resnet101_v1s
gluon_xception65
gluon_seresnext50_32x4d
gluon_senet154
gluon_inception_v3
gluon_resnet101_v1c
tf_inception_v3
tv_densenet121
tv_resnext50_32x4d
repvgg_b1g4
resnext26ts
ghostnet_100
crossvit_9_240
deit_base_patch16_384
rexnet_150
rexnet_130
resnetrs50
resnet50d
resnet50
resnetv2_50
resnetrs152
resnetrs101
resnet50_aa
resnet50_fastaa
resnet50_randaa
vgg19_bn
vgg16_bn
vgg13_bn
vgg11_bn
vgg11
vgg11_bn
vgg16
vgg19
resnet10t
resnet14t
darknet53
cs3darknet_m
cs3darknet_focus_m
cs3darknet_l
cs3darknet_focus_l
regnety_040
regnety_064
regnety_080
regnetv_040
regnetv_064
regnetz_040
```

`regnetz_040h`

**CIFAR-10 models** can be downloaded through (https://github.com/chenyaofo/pytorch-cifar-models). The training script for training trajectory experiment borrows from (https://github.com/kuangliu/pytorch-cifar).

```
VGG16
VGG13
ResNet34
ResNet50
ResNet101
ResNet152
ShuffleNetG2
ShuffleNetG3
PreActResNet18
PreActResNet34
PreActResNet50
PreActResNet101
densenet_cifar
DenseNet121
DenseNet169
DenseNet201
DenseNet161
ResNeXt29_8x64d
ResNeXt29_32x4d
MobileNetV2
RegNetX_200MF
DLA
DPN
PNASNetB
RegNetX_400MF
RegNetY_400MF
MobileNet
ResNet18
VGG11
SimpleDLA
ResNeXt29_4x64d
ResNeXt29_2x64d
EfficientNetB0
SENet18
ShuffleNetV2
GoogLeNet
DPN92
ResNet34-160
ResNet34-170
ShuffleNetG2-170
ShuffleNetG2-180
ShuffleNetG2-190
SimpleDLA-90
SimpleDLA-105
SimpleDLA-120
SimpleDLA-135
SimpleDLA-150
SENet18-60
SENet18-75
```

```
SENet18-90
SENet18-120
SENet18-135
SENet18-150
SENet18-165
ShuffleNetG2-170
ShuffleNetG2-180
ShuffleNetG2-190
densenet_cifar-135
densenet_cifar-150
densenet_cifar-165
MobileNetV2-135
MobileNetV2-150
MobileNetV2-165
LeNet
PreActResNet152
```

**WILDS models** (Camelyon17 and DomainNet) are trained by github (https://github.com/p-lambda/wilds).

**Zero-shot vision-language models** are provided in OpenCLIP (Ilharco et al., 2021). They are listed as follows in the pattern (`architecture`, `source`):.

(RN50, `openai`)
(RN50, `yfcc15m`)
(RN50, `cc12m`)
(RN50-quickgelu, `yfcc15m`)
(RN50-quickgelu, `cc12m`)
(RN101, `openai`)
(RN101, `yfcc15m`)
(RN101-quickgelu, `yfcc15m`)
(RN50×4, `openai`)
(RN50×16, `openai`)
(RN50×64, `openai`)
(ViT-B-32, `openai`)
(ViT-B-32, `laion400m_e32`)
(ViT-B-16, `openai`)
(ViT-B-16, `laion400m_e32`)
(ViT-L-14, `openai`)
(ViT-L-14, `laion2b_s32b_b82k`)
(ViT-L-14-336, `openai`)
(ViT-H-14, `laion2b_s32b_b79k`)
(ViT-g-14, `laion2b_s34b_b88k`)
(ViT-bigG-14, `laion2b_s39b_b160k`)
(convnext_base, `laion400m_s13b_b51k`)
(convnext_base_w, `laion_aesthetic_s13b_b82k`)
(convnext_base_w_320, `laion_aesthetic_s13b_b82k_augreg`)
(convnext_large_d, `laion2b_s26b_b102k_augreg`)
(convnext_large_d_320, `laion2b_s29b_b131k_ft_soup`)
(convnext_xxlarge, `laion2b_s34b_b82k_augreg`)
(ViT-B-32, `Model-B-32_Data-80M_Samples-34B_ lr-1e-3_bs-88k.pt`)
(ViT-B-16, `Model-B-16_Data-80M_Samples-34B_ lr-1e-3_bs-88k.pt`)
(ViT-L-14, `Model-L-14_Data-80M_Samples-34B_ lr-1e-3_bs-88k.pt`)
(ViT-B-32, `datacomp_m_s128m_b4k`)
(ViT-B-32, `commonpool_m_clip_s128m_b4k`)

```
(ViT-B-32, commonpool_m_laion_s128m_b4k)
(ViT-B-32, commonpool_m_image_s128m_b4k)
(ViT-B-32, commonpool_m_text_s128m_b4k)
(ViT-B-32, commonpool_m_basic_s128m_b4k)
(ViT-B-32, commonpool_m_s128m_b4k)
(ViT-B-32, datacomp_s_s13m_b4k)
(ViT-B-32, commonpool_s_clip_s13m_b4k)
(ViT-B-32, commonpool_s_laion_s13m_b4k)
(ViT-B-32, commonpool_s_image_s13m_b4k)
(ViT-B-32, commonpool_s_text_s13m_b4k)
(ViT-B-32, commonpool_s_basic_s13m_b4k)
(ViT-B-32, commonpool_s_s13m_b4k)
(ViT-B-16, datacomp_l_s1b_b8k)
(ViT-L-14, datacomp_xl_s13b_b90k)
(EVA01-g-14, laion400m_s11b_b41k)
(EVA01-g-14-plus, merged2b_s11b_b114k)
(EVA02-B-16, merged2b_s8b_b131k)
(EVA02-L-14, merged2b_s4b_b131k)
(EVA02-L-14-336, merged2b_s6b_b61k)
(EVA02-E-14, laion2b_s4b_b115k)
(EVA02-E-14-plus, laion2b_s9b_b144k)
(ViT-B-32-quickgelu, metaclip_fullcc)
(ViT-B-16-quickgelu, metaclip_fullcc)
(ViT-L-14-quickgelu, metaclip_fullcc)
(ViT-L-14-CLIPA-336, datacomp1b)
(ViT-H-14-CLIPA, datacomp1b)
(ViT-H-14-CLIPA-336, datacomp1b)
(ViT-bigG-14-CLIPA, datacomp1b)
(ViT-bigG-14-CLIPA-336, datacomp1b)
(ViT-B-16-SigLIP, webli)
(ViT-B-16-SigLIP-256, webli)
(ViT-L-16-SigLIP-384, webli)
(ViT-SO400M-14-SigLIP, webli)
(ViT-SO400M-14-SigLIP-384, webli)
(ViT-B-32-quickgelu, metaclip_fullcc)
(ViT-B-16-quickgelu, metaclip_fullcc)
(ViT-L-14-quickgelu, metaclip_fullcc)
```

BLIP (Li et al., 2022) and BLIP-2 (Li et al., 2023) models are from LAVIS library (https://github.com/salesforce/LAVIS/tree/main);

### A.1.3 Computation Resources

PyTorch version is 1.10.2+cu102. All experiments run on one 2080Ti and the CPU AMD Ryzen Threadripper 2950X 16-Core Processor.

| Models | ImageNet-V2-A | ImageNet-R | ObjectNet |
|--------|---------------|------------|-----------|
| 20% | 0.705 | 0.940 | 0.856 |
| 50% | 0.746 | 0.934 | 0.894 |
| 70% | 0.778 | 0.928 | 0.906 |
| 100% | 0.758 | 0.928 | 0.895 |

Table 5: Correlation strength of SoftmaxCorr with different number models. Weighted Kendall correlation ($\tau_w$) is used as the metric.

| Models | ImageNet-V2-A | ImageNet-R | ObjectNet |
|--------|---------------|------------|-----------|
| 20% | 0.540 | 0.841 | 0.786 |
| 50% | 0.542 | 0.877 | 0.846 |
| 70% | 0.579 | 0.867 | 0.846 |
| 100% | 0.597 | 0.884 | 0.849 |

Table 6: Correlation strength of MaxPred with different number models. Weighted Kendall correlation ($\tau_w$) is used as the metric.

### A.2   Sensitivity analysis on $\tau_w$

We use Scipy implementation of weighted Kendall's rank correlation $(\tau_w)$[1], which does not provide a p-value to indicate the statistical significance. It is because the null distribution of the statistic is unknown even in the additive hyperbolic case. Thus, to examine whether the evaluation metric is stable, we conduct a sensitivity analysis on $\tau_w$ by randomly dropping some models from the model pool. The results are summarized in Table 5 and Table 6. We find that $\tau_w$ has a consistent assessment on two methods with different numbers of models, suggesting the stability of $\tau_w$.

---

[1]https://docs.scipy.org/doc/scipy/reference/generated/scipy.stats.weightedtau.html

