# OpenReview forum: "What Does Softmax Probability Tell Us about Classifiers Ranking Across Diverse Test Conditions?"
_TMLR — Accepted by TMLR_

### Review · Reviewer_6Wtr · 2024-03-18

**Summary Of Contributions:**

This paper reveals the drawbacks of common uncertainty metrics related to Softmax prediction. In order to improve the measurement of model performance within both ID and OOD settings, a novel Softmax Correlation criterion is proposed. Specifically, it calculates the cosine similarity between a class-class correlation matrix which is constructed from the Softmax prediction matrix. Such a novel metric can capture both prediction uncertainty and diversity. Through quantitative and qualitative experiments, the effectiveness of SoftmaxCorr is validated on various OOD datasets.

**Audience:**

Yes

**Broader Impact Concerns:**

No ethical concerns.

**Claims And Evidence:**

Yes

**Requested Changes:**

Please see weaknesses for details.

**Strengths And Weaknesses:**

Strengths:
- This paper is well-written and logically organized.
- The experimental results are sufficiently good.

Weaknesses:
- The motivation is not clear. Why existing metrics are not good enough for OOD scenarios is not discussed in either the Abstract or Introduction. The authors just present such a novel metric and their motivation is unknown.
- Moreover, why the proposed SoftmaxCorr is designed this way is not explained. Why using the class-class correlation matrix C can help to capture both certainty and diversity is not justified. The proposed claims should have either theoretical or empirical evidence.
- Missing OOD generalization references:
  - Zhang et al., Can Subnetwork Structure be the Key to Out-of-Distribution Generalization? In ICML 2021.
  - Huang et al., Winning Prize Comes from Losing Tickets: Improve Invariant Learning by Exploring Variant Parameters for Out-of-Distribution Generalization. In arXiv 2023.
  - Foret et al., Sharpness-Aware Minimization for Efficiently Improving Generalization. In ICLR 2021.
  - Huang et al., Harnessing Out-Of-Distribution Examples via Augmenting Content and Style. ICLR 2023.

---

### Review · Reviewer_RkLL · 2024-03-26

**Summary Of Contributions:**

This work focuses on a setting where one is given various classifiers trained on some source domain and the goal is to rank them according to their performance on a target domain using only unlabeled data from the target domain.
It has been observed previously that a few quantities correlate reasonably with out-of-distribution (OOD) performance.
Examples of such quantities are in-distribution (ID) accuracy, average prediction confidence, and average confidence gap, where the latter two depend on predicted class probabilities (i.e., the softmax outputs) on test examples.

The authors of this submission aim to find a better way of using predicted class probabilities that goes beyond maximum or second maximum probabilities and incorporates “correlations” between predicted probabilities of different classes.
Let $P \in \mathbb{R}^{n\times k}$ be the matrix of predicted probabilities on $n$ test examples for a $k$-way classification tasks.
Let $C$ denote the uncentered class-class covariance matrix, $C = P^T P \in \mathbb{R}^{k \times k}$.
Let $R$ be a diagonal matrix where $R_{i,i}$ is the prevalence of the $i$-th class estimated using a strong vision-language model (e.g., a CLIP-like model).
The authors propose a new score called SoftmaxCorr that computes a similarity between $C$ and $R$ as follows:
$$
\mathrm{SoftmaxCorr} = \frac{\sum_{i,j} (CR)_{i,j}}{||C||_F ||R||_F}.
$$

With extensive experiments over many dataset pairs, models, and settings, the authors demonstrate that on-average SoftmaxCorr has the best rank correlation with OOD performance, outperforming baselines like average prediction confidence, average confidence gap, ID accuracy, and average thresholded confidence.

**Audience:**

Yes

**Claims And Evidence:**

Yes

**Requested Changes:**

1. This submission will benefit significantly from added experiments that compare SoftmaxCorr and a simple baseline that estimates OOD accuracy by treating the reference model predictions as ground truth.
2. Justifying SoftmaxCorr definition better (please see my comments above).

**Minor improvements**
* In the introduction, “...there is a pressing need to assess ML model performance in unlabeled testing environments where traditional evaluation metrics may prove inadequate.” I suggest clarifying which traditional evaluation metrics are referred to here. The traditional evaluation metrics I think of require labels.
* I suggest adding (Dzuigate et al., 2020) to the list of references where the goal is to predict in-distribution generalization.
* In “0.949, 0.627, 0.934, 0.810, and 0.857” the number 0.934 does match with Table 1. Please correct either the sentence or Table 1.
* “well correlate” → “correlate well”
* In the abstract: “matrix–constructed”  → “matrix constructed”
* In the related work: “this work investigate” → “this work investigates”
* In the related work, “There are few works consider the characterization of model OOD generalization”. A “that" is missing. Furthermore, it is not true that few works consider bounds for OOD generalization.
* “Machine learning models should generalize from training distribution to OOD dataset.” The papers cited in the context of this sentence seem to be not directly related. I don’t think this sentence needs citations. If one decides to cite, there are 100s of papers that should be cited here.
* In the related work, “This work investigate a different task from unsupervised accuracy estimation.”. One can argue that unsupervised accuracy estimation is very close to the ranking problem, because any method of estimation is also a method of ranking.
* The abbreviations “IntraCorr” and “InterCorr” are never used.
* The notation for source examples is never used and can be removed.
* Section 4.1, “Paradoxically, identifying the failure patterns itself requires labeled OOD data.”. Could you please add citation(s) for this claim?
* Section 5.2, “On OOD sets of Camelyon17, all methods correlate weakly with OOD performance, which may be caused by under-specification (Wenzel et al., 2022).“ This message is already conveyed in the paragraph above.


**Related work**

[1] Dziugaite GK, Drouin A, Neal B, Rajkumar N, Caballero E, Wang L, Mitliagkas I, Roy DM. In search of robust measures of generalization. Advances in Neural Information Processing Systems. 2020;33:11723-33.

**Strengths And Weaknesses:**

**Strengths.** This submission presents an extensive set of experiments over many datasets, models, settings, and OOD performance ranking baselines. Furthermore, rank correlations are computed in a few settings: (a) both model and ID/OOD datasets change, (b) datasets change but model is fixed, and (c) datasets are fixed but model changes. The central claim that SoftmaxCorr is better correlated with OOD performance on average is well-supported. To my best understanding, there are enough details to reproduce experiments.


**Weakness #1: Insufficient justification for SoftmaxCorr definition.** I find that SoftmaxCorr definition is not sufficiently motivated and expanded upon. Below are a few questions and concerns about it.
* In Section 4.3, the “class-class correlation matrix” $C$ is defined as $P^T P$. There should be a $1/n$ factor to counter the growth with $n$.
* Note that $P^T P$ is not a correlation matrix. It is not even a covariance matrix, because columns are not centered. Why is $C$ defined like this, but not like a correlation matrix?
* Why is the reference matrix $R$ picked to be diagonal? Why is it not defined like $C$ but for the reference model?
* I am not familiar with the cosine similarity between two matrices in the way the authors write:
$$
\mathrm{cos}(C,F) = \frac{\sum\_{i,j} (CR)\_{i,j}}{||C||\_F ||R||\_F}.
$$
I have seen definitions that flatten the matrices into vectors and then compute cosine similarity. That looks like this:
$$
\mathrm{cos}(C,F) = \frac{\sum\_{i,j} C\_{i,j} R\_{i,j}}{||C||\_F ||R||\_F},
$$
and shows up in other areas of ML (e.g., measuring alignment between two kernel matrices). Please elaborate on your definition of cosine similarity. What range of values does it take? For a fixed $R$, which $C$ maximizes it? Furthermore, since $R$ is picked to be diagonal, I suggest simplifying SoftmaxCorr definition instead of stating it for a generic $R$.

**Weakness $2: Reliance on a strong reference model.** SoftmaxCorr relies on a strong reference model to estimate class prevalences reliably. It is likely that this gives a significant advantage to SoftmaxCorr over other baselines that do not use such a reference model. If such strong reference models are allowed, a natural way to estimate OOD accuracy is to compute disagreement between main model predictions and reference model predictions (i.e., treating reference model predictions as ground truth).


**Weakness #3: There are many vague statements [minor].**
* In the introduction, there is a sentence “We thus propose SoftmaxCorr is a prediction probability-based metric that quantifies the extent to which class predictions made by classifiers get confused with each other“. I do not see how this is exact in a technical sense. I suggest adding a small paragraph in later sections showing this connection.
* Throughout the work, the authors refer to $C$ as a correlation matrix, which it is not. Furthermore, there are places where it is treated like a confusion matrix, which it is also not.
* In Section 4.3, “A model scoring high on the SoftmaxCorr measure excels in two critical parts: 1) It demonstrates a high level of prediction certainty, ensuring that its intra-class correlations are strong and reliable. 2) It achieves a broad prediction diversity, indicated by the alignment of the diagonal elements in its class correlation matrix with the class marginal distribution.” These two points are unclear to me. I think it would be nice to expand SoftmaxCorr definition and show explicitly how these two effects arise.
* In Section 5.5, “(2) the euclidean distance between diagonal elements in the class correlation matrix and estimated class distribution (Diversity). It measures whether models make diverse predictions whose distribution matches the estimated distribution of each class.” How is this interpreted as diversity?

---

### Review · Reviewer_wfZb · 2024-04-02

**Summary Of Contributions:**

This paper proposes a method (SoftmaxCorr) to rank the performance of classifiers on unlabeled out-of-distribution data. Experiments on various datasets demonstrate the effectiveness of SoftmaxCorr.

**Audience:**

No

**Broader Impact Concerns:**

This work does not present direct ethical concerns or adverse broader societal impacts.

**Claims And Evidence:**

No

**Requested Changes:**

1. Figure 1 should be more clear. For example, the way to obtain the different classifiers should be clarified (using different network structures or different training methods?)  How many types of VLMs have been tested? What is the meaning of the line with shadow?
2. The equation on Section 4.3 is wrong $C_{i,j}=\sum_{n=1}^NP_{i,n}P_{n,j}$, instead of $C_{i,j}=\sum_{n=1}^NP_{n,i}P_{n,j}$.
3. What is the relationship between $\mathbf{R}_K$ and $\mathbf{R}$?
4. The advantages of introducing SoftGap compared with MaxPred should be discussed in the paper.
5. Missing the sensitive test with the evaluation metric Kendall’s rank correlation $\tau_w$.

**Strengths And Weaknesses:**

Strengths:
1. This paper proposes a method to rank the performance of classifiers on OOD data without using annotation.
2. A lot of empirical results demonstrate the effectiveness of the proposed method.

Weaknesses:

1. Though the paper proposes SoftmaxCorr, its advantages compared with previous methods are unclear. Why is SoftmaxCorr better than previous methods? How does the proposed method utilize the two characteristics described in the paper  (Prediction certainty and Prediction diversity)?
2. The writing of the paper is poor. See the Requested Changes.

---

### Decision · Action_Editor_ebrX · 2024-05-06

**Recommendation:** Accept as is

**Comment:**

In reviews, the reviewers raise concerns about typos in equations, evaluation metrics, motivation, and assumptions about requiring a strong inference model. After the authors' response, most concerns have been addressed. All reviewers vote for leaning accept.

By considering that how to properly select the model in the OOD setting and the experiments can be valuable for future research. We decide to accept this paper with minor revision. The authors should follow the advice of  reviewers to further improve the motivation of this paper.

**Audience:**

This paper targets the problem of how to select a good model in an OOD setting without a validation set. This question is important and may be of interest to those doing research related to OOD detection and model selection.

**Claims And Evidence:**

This paper proposes a method to rank the performance of classifiers on OOD data without using annotation. Most reviewers agree that the results are accurate and convincing. Major claims are well supported.